# Being with a Puppet: Literacy through Experiencing Puppetry and Drama with Young Children

Olivia Karaolis

School of Education and Social Work, University of Sydney, Sydney, NSW 2006, Australia;
olivia.karaolis@sydney.edu.au

**Abstract:** Puppets have a long association with early childhood education and have played a much-loved role in children's learning and development. This paper tells the research story that investigated how the magical creature of a puppet facilitated connection, play, communication, and engagement with children who experience disability. We discovered how puppets can be combined with drama approaches and utilized in group activities for enabling literacy development by early childhood educators. In being with a puppet, adults found new 'ways' of supporting all children's interest, meaning making, and contribution to group learning experiences. Puppets were found to invite children into conversations, to encourage their expression and creativity, in a way that was uniquely their own. Educators found that being with a puppet supported their relationship with the children to one that was more playful and positive as it altered their perception of the children and their ability to recognize their non-verbal communication.

**Keywords:** puppetry; inclusion; early childhood; literacy

## 1. Introduction and Background

Puppets can play a special role in the lives of young children. Just like a favorite toy or doll, a puppet can take on any form (human, animal, or other creature) and be used to play, comfort, and make or express meaning [1]. A puppet is simply defined is an object, that can be brought to life through movement and voice by the puppeteer. Examples include marionettes, finger puppets, hand puppets, and puppets made from paper, cloth, or other found materials. In early childhood education, the focus of this study, puppets have been used to for children's play, social and emotional development, language, creativity, early literacy development [2–5], and to support inclusive practice and a sense of belonging [2].

Belonging was seen to be developed through the three-way relationship between educators, children, and the puppet [2,4]. In this relationship, the power dynamics and quality of communication were seen to change. Educators utilizing a puppet became more playful, less threatening, and able to easily form connections with the children. In puppet play, the children were more willing to share their interests and strengths, aspects of themselves that may otherwise not have been revealed. Educators faced with the tension between play-based learning that is co-constructed with children and the pressure of school readiness may find the puppet invaluable in planning for curriculum that meets the obligations of standardized learning outcomes and the uniqueness of every child [6,7]. One such curriculum area is literacy.

The definition of literacy in this study is consistent with the Early Years Learning Framework [8] (EYLF) (2000) and one that views literacy as the "capacity to use language in all its forms" ([8] p. 41) [9,10]. Forms of language include a vast range of texts or modalities including books, songs, music, movement, drama, improvisations, sound, dance, puppetry, and other artforms. This complex or expansive view of literacy is described by Eisner ([11] p. 5) as "a way to be in the world, another way to form experience, another way to recover and express meaning". His interpretation moves beyond the simple or traditional view

of literacy as being about reading books and to one of being able to form meaning from a range of objects and experiences, including the creative arts. Marie Clay's work adds to this understanding of early literacy that evolves differently for each and every child [12]. Educators of children in the early years play a central role in supporting children to become literate through understanding their unique literacy development, by their pedagogical choices, conversations with children, relationships, and provision and selection of texts [12].

Puppets have been used as a text in several studies that have employed a more traditional approach to literacy [13,14], as well as in research that considered literary more broadly [1,15,16]. Remer and Tzuriel [16] described how the puppet was shown to support the quality of the interactions between children and adults, bringing about a change in their tone and quality, allowing both participants to reveal new parts of themselves and to learn more about one another. Alchrona [4] attributes these changes to the three-way relationships that result from the presence of the puppet and the perception by the children of the puppet as real. In her study, this investment in the emotional life of the puppet supported the children's engagement. For example, the preschool aged children developed the character of Hedvig, her puppet, through asking questions about her favorite song or birthday celebration ([4] p. 176).

Playing with puppets develops children language and literacy skills beyond direct instruction [17]. For example, children may use finger puppets or small hand puppets to act out situations and play with words and sounds; in doing so, they are expanding their vocabulary and pragmatic language use [17]. Educators can use puppets as props in a range of dramatic play situations that help them to construct knowledge of the meaning of words. In addition to expanding vocabulary, puppets support children's comprehension and active listening [16,17] by retelling aspects of the story to a puppet or taking on roles of the characters to create alternative endings and inhabit the text. In translating or interpreting the text, relationships between the children and their teachers are nurtured, adding to the wellbeing of all participants [16].

In my own research and practice [2] my first intention was to build the relationship between the children and the puppet. For example, the children were often involved in the naming of the visiting puppets and in building its personal story, qualities, and interests. They would ask questions about puppets that had visited them on other occasions, draw them pictures and ask to know their favorite game or share their favorite picture book, all of which we would weave into our next session. The children and the puppet were building a group in which they all had equal membership [12]. For educators, a puppet was seen as invaluable for bringing all the children into whole group learning experiences and to foster the conversations essential for literacy [10], for learning about the children's prior knowledge [6], and supporting ALL children to develop their capacity to express themselves and contribute to their community [12,17].

My emphasis on the word ALL is intentional, as it refers to every child, including those who experience disability, children learning English as an additional language or dialect, and children from diverse cultural or family backgrounds. Just like the concept of literacy, there are many ways of describing or talking about the notion of disability, all of which reflect our views and values. Our idea of disability is one that is shaped by a myriad of beliefs and for the most part, a perception of disability as being different and different in a way that was inferior to or outside the 'norm' [18]. This view is related to the medical model of disability [19] and founded on a view of disability as an ailment or impairment, one that is based on a biological condition that has damaged the individual. From this perspective, individuals with disability are seen as 'less' than those that do not experience disability, with problems that need to be solved or needs that require the attention and intervention of others [19,20]. In education, such labels can lead to segregation of some students into special classes, units, or learning groups, to be defined by their disability and subject to learning expectations that are less than others [18].

The social model of disability considers disability differently and looks beyond the individual with as disability as a 'problem'. Rather, it considers the problems that individuals

with disability have imposed on them by social attitudes or physical barriers [19]. Examples of this may include a lack of access to buildings or public spaces, unwelcoming educational settings, assumptions of competence, and social exclusion. Barriers to literacy learning may be an over reliance on spoken language, assumptions about children's ability, environments in which young children do not experience a sense of belonging, or because of limitations to act with agency and autonomy [19,20]. These barriers can be reduced through a design of literacy experiences that appeal to a range of interests and offer multiple possibilities of expression [18].

This article will introduce the possibilities of puppets in early literacy development and describe their potential to support children to share their voice and make meaning through verbal, nonverbal, and relational communicative acts [4]. It draws on a study [2] that led educators to change the way they viewed, interacted, planned for, and valued the contribution of every child and supported their literacy in the richest sense [9].

## 2. Materials and Methods

I chose portraiture as the research methodology, an approach that includes the following "dimensions—centering relationships, seeking goodness, and attending to the aesthetics of storytelling" [21,22]. Lawrence-Lightfoot and Davis [21] define goodness as the focus on the positive and to "resist this tradition-laden effort to document failure" (p. 50). Their sentiment appealed to me as I wanted to resist a tradition, that is the tradition of depicting disability as a deficit and to capture the strength and potential of every child. The dimensions that underpinned portraiture were aligned with the overarching aim of my study to share stories of how drama and puppetry can support the communication and self-expression of every child [20]. The methodology, created as part of Sarah Lawrence-Lightfoot's [21] investigation of high schools in the United States, and used widely to explore other educational settings, allows the researcher to observe the subjects of the study with the detail of a scientist and explore the multiple perspectives present in every situation [21,22]. The methodology also focuses attention to what the participants perceive as "good", and to contrast what the children see as "good" in their day-to-day experience and to compare that with the adults and across the three different research settings [22]. Portraiture has been used as a research methodology to answer a range of research questions, including the strengths of young adult literacy learners [23]. It is a strength- and place-based approach that seeks to understand those involved in their study and honor their experience [23]. To translate the portraits to an article this size, I have selected vignettes or moments from the 'portraits' in my original study to capture an example of a reoccurring finding or highlight a common theme [24].

The study received ethics approval from the University of Sydney and written consent obtained from educators, parents, and caregivers prior to the data collection. Opportunities for children to give their assent was ongoing and all children were invited to participate in the research activities and express their agreement through actions or words. It should be noted that on some occasions, a puppet was utilized to make the request. All participant names and other identifiable information were changed to protect the identity of those involved in the study.

My study involved over 60 children, between the ages four and five, and nine educators (eight female and one male) at three different preschools located in urban areas of Sydney. I had a relationship with two of the three preschools in my role as a Tertiary Mentor at the University of Sydney and approached the centers as I was aware that the programs included children with disabilities. The third center was managed by the local council and provided me with three different types of preschools, private, government, and not for profit. The center director then invited room leaders to take part in the study and all expressed an interest in the possibility of learning more about children with disabilities in their care. My role is best described as teacher/researcher implementing the research activities at each of the preschools. The research activities were co-designed with the educators, and different drama strategies were discussed and then selected to 'answer' the

research question of "Do drama processes act as facilitators to include all in early childhood settings? If so, how?".

To research "how" drama and puppetry can support the communication and self-expression of every child this with accuracy and insight, I needed to immerse myself in the life of each of the preschools. So, I became a weekly visitor, observing the children, meeting with their teachers and the center director, and being with the children in a range of learning experiences, along with a puppet of course. The research period was for 10 weeks at each setting and involved a ninety-minute visit at each program. This time allowed for observation, puppet and drama activities, and interviews with the participants.

Data from participant educators were obtained through a variety of methods that included semi-structured interviews conducted after the drama workshops, written reflections, images, children's drawings, and informal conversations recorded in the researcher's journal. At the end of the research period and drama intervention, all educators were asked to respond to a series of questions in person and via email to describe their perception of the children's engagement and participation in response to the drama experiences. The questions were the same for every preschool and included:

1. Which forms of drama, if any, engaged all the children in a group learning experience?
2. Have you observed any changes in the children as a result of participating in drama?
3. Will you use include any of the drama experiences in your curriculum? If so, which ones?
4. Did you feel that collaboration and co-teaching was helpful to your teaching practice? If so, what was the most valuable aspect of this partnership?
5. Describe any changes you may have felt in your understanding about drama as a way of teaching from participating in the research.

This information was used for analysis alongside my own research journal. The views of the children were elicited through open-ended questions and drawings; for example, children were invited to "draw" a picture about the part of the drama they liked best (Qu.1) or shown an image of them with the puppet and asked to "Tell me about this picture" (Qu.2) or shown a picture and asked to describe their feelings (Qu.1). Once this information was gathered it was coded using In Vivo Coding and then Dramaturgical Coding [25]. The codes were then formed into themes to capture the main findings or research topics, with puppets appearing frequently in the analysis. Other themes to emerge were engagement, motivation, and communication. Their relevance for development of emergent literacy is discussed below.

## 3. Being with a Puppet

The Early Years Learning framework ([8] p. 13) tells us "there are many ways of living, being and knowing". In this study, a puppet was shown to be one of those ways enabling educators with a tool to support relationships, participation in group learning experiences and to engage in literacy and imaginative experiences. The puppet allowed the child and the adult to shift from doing to being through the depth of the encounter and the combination of the real context and the imagined or fictional context created with the puppets. In these interactions, the children were able to learn more about one another in a space that communicated difference as positive and natural. The interactions with the puppet supported the children, and in some instances the adults, to explore and appreciate differences. The encounters with the puppet were also valuable in challenging some perceived or developing ideas about some of the children, their capabilities, their competence, or their development. One such research vignette is the story of Ben and his role, with the rest of the children in a literacy rich experience as portrayed below. I chose this vignette to highlight the "miniature of a more significant picture" of what small acts can tell us about a child [23,24], the miniature in this case being the value of the creative arts.

Literacy was a very important area of focus at this center and an area that the teachers felt not all the children were enjoying or interested in. The teachers requested to continue an exploration of books through drama and puppetry. Rather than working on a familiar

book, I thought it might be worth introducing a new story, one that shared the name of one of the children in the title. My hope was this may entice Ben, a very quiet boy who often withdrew from group experiences to wander towards the walls of the classroom, his fingers tracing the bricks on the fireplace as he skirted around the room. Clay's theory was illuminating, as at in our initial few sessions, I asked nothing of Ben and interpret his actions as "roaming" [12]. I valued the chance to observe his familiar habits, that allowed us both the space to know one other. This was of particular importance as all three teachers had expressed their concerns about his behviour. When he was playing outside, Ben looked uncertain, he would play chase only if the boys ran after him, then stop and lean on a pole, sucking his thumb and watching as the boys ran around him. At other times, he would sit on a step by his teacher and stare into space. I wondered what he was thinking. The snapshot below was one piece of information used to "see" Ben without too much attention to the deficits described [23,24] and speak to both "the head and the heart" [23].

*3.1. Vignette 1*

It was a cool day and I sat in a sunny spot on the patio steps. Lyndall, a very chatty four-year-old ran over the moment she saw me to check that I had Mabel. Joel drove by in a red car and gave me a big smile. After a five-minute warning, Miss Jola called the children inside and the little folk bounced to the mat. Making a circle was much easier this time and Miss Jola, the room leader started the day with Welcome to Country. As she told the children that I have come to visit them, she was interrupted by Lyndall, who called out, "Mabel too". I moved down to the floor and gesture to Joel to sit next to me. Without hesitation he came right over. I then say:

*Olivia: Hello beautiful ones, I am so happy to see you again today. I wanted to ask you...I am having a teensy bit of a problem waking Mabel up today. (I gesture towards Mabel, sleeping in the basket and get her out, cradling her in my arms) Mabel, wake up...". (Mabel continues to snore) I think we might need a magic word. Does anyone know a magic word?*

*Henry: (one of the taller boys in the group who loves ninja turtles offers) Abracadabra*

*Olivia: Ok, Henry, thank-you... everyone let's try it"*

*(All the children say "Abracadabra" or a word close to it. Jack smiles and moves his feet together as Mabel snores)*

*Olivia: Mmm... I think we might need another word. (At that very moment Jack makes a vocalisation that sounds like, "theee")*

*Olivia: Oh, thank you Jack, let's try that word...everyone...theee. Some of the children jumped in and repeat "theee". I noticed a few of the older children looked a little dubious. This is quite natural as it does not sound like a typical 'word', even the teachers needed a minute to catch on, with one of them repeating quizzically, "Three"? I quickly hopped in and said how much I liked the word. Jack smiles and bangs his feet together again as Mabel woke up...*

*Olivia: Thanks Jack...Good morning, Mabel, did you sleep well? (Mabel nods). Look where we are...*

*Mabel: Oh...Hi everyone...it is so good to be here Olivia, I am so happy... (Mabel jumped up and down, very, very excitedly) Her jumping created a stir, the children replied by saying, hello or with a wave or a giggle. All eyes were watching Mabel and a lot of children moved forward on their knees to be closer to her. Another teacher needed to remind the children to stay on their bottoms so that, "all the friends can see".*

*Olivia: (to the children) Do you remember Mabel's favorite game? I think she is going to want to play it again today'.*

*Mabel: I do, I do... (She is so excited to see the children). I reminded the children that we needed to be in a circle, as the excitement of Mabel has brought us all forward in a group.*

*With a bit of help from the other teachers, our circle forms again and we pass a 'hello' around the group. I noticed Jack looked at the boy next to him with a huge grin and Joel was following all of the actions, including making a silly face. He turned his body slightly away from the child to his left, his teacher walked over to him about to adjust his body. I was too quick and gave my head a tiny shake and mouth, "He is ok". I suspected this was his way of managing all stimuli that was going on around him. It made perfect sense to me as it seemed he was reducing some of the sensory information in his environment. His teacher moved back to the computer and I also suspect that I have offended her...*

*But we move on and pass a squeeze. It was fun to see how the children waited to receive a squeeze from their friend before they squeezed the person next to them. A few children needed a reminder to squeeze, and a few random squeezes started up, I decided it was time to move on to the next activity and moved back to the sofa from the circle as Mabel asked me to read her favorite book, Ben and the Beast.*

*Hearing his name in the title caused the exact response that I had hoped for, Ben, after recognizing his name, hopped up and sat right at my feet. Some of the other children smiled as they too recognized his name. The story was so easy to bring to life with actions. A tale of the simple quest of a young girl. Along with the heroine and Mabel, we walked through the forest, scrunching the leaves under our feet. We shivered in the dark woods and took on different characters as they entered the story, a rabbit, a mouse, and a snake. We stopped to eat the donuts that Ben, our hero, stashed in her hair and made soup with the beast before defeating him and sending him rolling down the hill. The book was quite long, and I was thrilled that all the children stayed for the entire shared reading expressed their understanding through role play and movement. Ben's response was described by Miss Belinda in the email below:*

*Jola and I thought it went really well! That was a long book, and the children showed great interest. Ben enjoyed the interactions throughout the book, and it was very interesting to see how the children attached to different characters. I look forward to our afternoon session!*

*I also sat the children in a circle following our activities, and as Jack is very into incy wincy, we all sang it together and passed the spider to each other. Jack was interacting so well in this experience.*

### 3.2. Puppets and a Space for Being

The description above illustrates three ways that puppets and drama can contribute to children's membership in early literacy practices. The first of these is the power of the puppet to engage children in language rich experiences. Research shows the value of talking and listening to children for both the development of spoken language and to provide a foundation for literacy [10,26,27]. Flynn [28] describes the value of routine conversations to provide children with regular and informal opportunities to participate in conversations. The weekly visits from a puppet created this routine for the children as our workshop began with the introduction of a puppet 'friend'. The puppet was selected with intention and to frame our conversation. For example, the puppet play with Mabel was designed to motivate the children to participate in the literacy experiences and be part of exploring a new text. The children were involved in group discussions and encouraged to have individual conversations, ask questions, and participate in drama with spoken and non-spoken language. They were introduced to new vocabulary, practiced listening to one another, and pragmatic language skills in improvisation, movement, and drama games—all with the scaffolding provided by the puppet [12]. These creative experiences elicited the children's self-expression and supported them to make meaning of their world and the world of others as they were motivated by the presence and the act of being with the puppet [3,4].

The initial interaction with Mabel guided or modeled for the children how to communicate and be with others. In the outline above, the children were seen to connect what

they knew about the situations in the story and build on each other's knowledge. They were developing their ability to make meaning of the world around them and to have their contributions valued. In helping Mabel, the children were able to act on her behalf and expand their understanding of feelings and the feelings of others. The puppet added to the benefits of drama, that "can evoke thoughts and feelings that invite us to wonder, to move across the boundaries between what we know and what we might yet know, and to change our actions [29]. When they connect our hearts, souls, and minds to those of others, and allow our lives to interact in all their wholeness, they can be transformative" (p. 258).

In this study [2], puppets were transformative in two distinct ways; they led to the transformation of the actions of the children, such as Ben's sense of belonging, and the transformation of the educator attitudes about the children. One participant educator observed:

"The use of puppets really gets the children's attention and then asking them to emulate a specific character/animal enables the children to all engage in their own way. It was amazing to see increased engagement in many of the children and even more amazing to see how each one of them reacted/acted differently. I cannot wait to start incorporating these into my wrap up group times and think the puppets would help enhance the experience for all the children in our room" (Kitty personal correspondence 4/12/2019).

The puppet gave the educators insight about the children, another perspective from which to view them respond to novel situations., situations in which they were emotionally engaged and motivated to act with agency and intention. The puppet gave the children the impetus to try new ways of being and for the educators to explore new ways of being with the children. The puppet granted them permission.

## 4. Puppets for Being with the Children

*"He is all over the place—socially and emotionally. I worry about his language, poor eye contact, very poor ability to concentrate and his body coordination is not that great either. The gap between his peers is widening every week" (personal communication, 5/5/2019)*

The puppet, through changing the interactions between adults and children, created the conditions to be in the present, for being and "engaging with life's joys and complexities, and meeting challenges in everyday life" ([8] p. 7). Puppets changed the quality of interactions in several ways and created a relaxed environment as the puppet was playful and non-threatening. As seen in other studies [4,16,17,30,31], the traditional power dynamic between adult and child altered with a puppet as the adults were given an opportunity to play with the children when co-developing the puppet character. This changed the nature of the learning experiences as meaning was constructed jointly and not provided solely by the adult. The role of the adult was much more of a co-player in the construction of the puppet's identity and the puppet story or play. Through dramatic experiences with a puppet, the children were more engaged in group learning experiences and played with language and sounds in the creation of the puppet's identity [4].

In learning about the puppet, children were able to share their own stories, their interests, and ideas. The puppet gave the children new ways to be together and express their thoughts through words, movements, and play. In observing the puppet play, the educators gained insight and another way to connect to the children, to see what they were able to do, and appreciate their individual and authentic responses. The focus was less on what the children were unable to communicate, as these expectations were removed, and the children were given a place to be themselves and respond to the sensory creature of a puppet. The quote above is worthy of analysis as it provides an example of how one educator viewed a child prior to the research. Her focus was on the assumed deficits, the ways in which this child was a 'lesser' version of the other children, because he was not meeting certain milestones or learning in the same way as she expected. Something shifted after she observed Ben interact with the puppet, for she could see that in the right or different circumstances, he could speak, follow the play, and respond with thoughtfulness and a recognition of the feelings of other children. All of these actions were surprising for

the educators, surprising in a good way as it prompted their thinking about the many ways children are literate. Ben needed more confidence in participating and that was brought about by the object of the puppet, it motivated Ben to join the group and anchored his attention. We wondered if he would able to understand the activity because of the visual tool of the puppet? Was it the calming effect of having the chance to meet the puppet individually? Would he benefit from being in a smaller group? The puppet allowed us a way to see more of Ben and the other children strengths and for teachers to consider how to adjust the learning environment and support learning and wellbeing through drama and puppetry. As one educator wrote:

> *"I loved how you've used our concerns over the last week (children's emotional regulation and challenging behaviour) to guide the puppets interactions with the children. The cuddles, the gentle hands and the feelings chair were all wonderful ways for children to revisit these issues away from literature and discussions. Providing them with an outlet and encouraging them to use this safe space enables their sense of belonging and wellbeing within the room." (Personal correspondence, 5/17/2019).*

## 5. Conclusions: Puppets to See "Goodness"

Puppets were shown to promote and sustain engagement in small and large group early literacy learning experiences in all three groups [31]. The puppets were also seen to promote the children's communication and confidence in all preschools and support the conversations between children and adults. Consistently, the puppets acted as a text and just like a book, provided children with a starting place to explore story and play. The puppet, in its presence alone was seen to support children to communicate and make meaning of their world without preferencing spoken language and in this way, played a key role in bringing about the participation of all children in this research that respected their linguistic diversity and unique selves. The puppet was accessible as the children could make meaning through the sense of sight, touch, and sound; this helped the children connect to the puppeteer and brought about relationships and a sense of belonging. Children were able to express their ideas in group situations with the puppet that were both exciting, safe, and shifted the perception of the children in the eyes of the adults. By being with a puppet, children and adults celebrated one another and most importantly, the gifts of every child. The puppets evoked the "goodness" [22,23], illuminating the strengths and prior knowledge, intentions, and emerging literacy of the children. Drama processes when combined with puppets were powerful in facilitating the inclusion of all children in these early childhood settings for scaffolding purposes and as an ideal support "for roaming around the unknown" [12]. This study contributes to our understanding of how these magical creatures support our learning and enrich the relationships and play so essential to early literacy development. Through puppets, we can reframe our thinking and evaluate our assumptions. The puppet is the perfect primer for early literacy development that encompasses the whole child [30,31]

**Funding:** This research received no external funding.

**Institutional Review Board Statement:** Not applicable.

**Informed Consent Statement:** Not applicable.

**Data Availability Statement:** Not applicable.

**Conflicts of Interest:** The author declares no conflict of interest.

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
