# Peer review of "Being with a Puppet: Literacy through Experiencing Puppetry and Drama with Young Children"

_education, doi:10.3390/educsci13030291_

Round 1
Reviewer 1 Report
Lines 101, 102, 103:
“I chose portraiture as the research methodology, an approach that includes the following “dimensions - centering relationships, seeking goodness, and attending to the aesthetics of storytelling” (20).
Lawrence- Lightfoot and Davis (22) define goodness as the focus on the positive and to “resist this tradition-laden effort to document failure” (p. 50).”
This seems to be a quite complex construct and readers might not be familiar with this approach. I would suggest expanding a bit more.
As for the concept “well-being”, authors should look at the research carried out by Professor Ferre-Laevers and at the Leuven Involvement Scale-, at it seems more suitable to the present paper
The chosen instrument seems to be the main pitfall of the study, as it ws originally developed to High School students. As a consequence, I wonder if the instrument by Ferre-Laevers wouldn’t be more suitable?
The study is quite interesting and valuable, I could not agree more that “to capture the strength and potential of every child” is the key of every study.
There are other aspects which might be improved in the theoretical backgound, fos instance, when authors talk about ‘literacy’ they should look and cite the work of Mary Clay, a stepping stone to emergent and litercy development studies. The studies from Rebecca Treiman are also worth a ‘sneak-peak’. I firmly beliebe if authors take in these advice, they will build a stronger paper.
The results could be put together in a Table together with a proper category.
I hope authors take this comments into consideration.
Looking forward to read the revised version.

Author Response
Dear Reviewer,
Thank you for your thoughtful feedback. It is very welcome and helpful in bringing this research story to life. I have responded to your feedback below and in the body of the article.
The chosen instrument seems to be the main pitfall of the study, as it ws originally developed to High School students. As a consequence, I wonder if the instrument by Ferre-Laevers wouldn’t be more suitable?
My choice of portraiture as a methodology was to answer one of my intentions as a researcher and that being to create a study that would be accessible. The use of "stories" (Bruhn & Jimenez (2020) of portraits as an educational approach has been used widely in educational research, nursing and critical theory as they allow the researcher to capture and tell of the unique experience of the participants in the study and engage the audience in the research question and findings (Faulkner et al., 2022) As the audience for this study was Early Childhood Educators, an approach was needed that would both involve them and resonate with their own teaching and work with children. In writing this response to you, I see how this justification for the choice of instrument was needed in the article and have developed this further- thank you.
The study is quite interesting and valuable, I could not agree more that “to capture the strength and potential of every child” is the key of every study.
There are other aspects which might be improved in the theoretical backgound, fos instance, when authors talk about ‘literacy’ they should look and cite the work of Mary Clay, a stepping stone to emergent and litercy development studies. The studies from Rebecca Treiman are also worth a ‘sneak-peak’. I firmly beliebe if authors take in these advice, they will build a stronger paper.
Thank you for Marie Clay- yes her concept of "resources" and the definition of emergent literacy added to the discussion and my appreciation of the effect of the puppet in revealing this to educators- another big "thank you"
The results could be put together in a Table together with a proper category.
I agree this may add clarity and yet resist due to the methodology and its focus on description and qualitative information.
Thank you again- I hope that you see the influence of your work.
I am so very grateful.

Reviewer 2 Report
There are not significant novelty in the article. The information are generic about the experiencing puppetry 2 with young children recommend for Book Chapter with additional information about current trends , not considerable as a research article as per journal repute in current form. Lot of research has been available in this area from last 10 years. Make a comparative analysis and justify the author's model with existing research articles and justify the novelty of the article.
Author Response
Thank you for your feedback. I have addressed how the findings are novel and build on the existing literature. I hope this is evident in the analysis of the vignette.
Thank you again for your suggestions.
Warmly,
Olivia
